# Impact of Philosophy for Children and Its Challenges: A Systematic Review

**DOI:** 10.3390/children9111671

**Published:** 2022-10-31

**Authors:** Mohd Kaziman Ab Wahab, Hafizhah Zulkifli, Khadijah Abdul Razak

**Affiliations:** Centre of Diversity Education, Universiti Kebangsaan Malaysia, Bangi 43600, Malaysia

**Keywords:** philosophy for children (P4C), the community of inquiry (COI), non-cognitive, higher order thinking skills (HOTS), challenges

## Abstract

Philosophy for children (P4C) has been implemented worldwide. P4C has been researched empirically in order to evaluate its effectiveness and address the current lack of a systemic literature review of research on P4C. Therefore, this SLR study aims to identify how P4C positively affects aspects other than students’ thinking and the challenges that teachers and students face in implementing the program. The methodology and writing method used was PRISMA (preferred reporting items for systematic review and meta-analysis). Articles and materials related to the topic were located primarily using two databases, Web of Science and Scopus. Using thematic analysis, this SLR derived five main themes, namely (1) higher-order thinking skills (HOTS), (2) safe environments, (3) civilized students, (4) democracy in discussion, and (5) the culture of thinking in the classroom. There are also challenges faced by teachers and students.

## 1. Introduction

Many have agreed with Dewey’s rallying cry that ‘Education should be about developing children’s thinking, not by tally them what to think but by helping them to find their own path to meaning’ [Fisher 2013, p. 5]. One program that can help children to find their own path towards meaning is the Philosophy for Children (P4C) program. This programmed was introduced by Mathew Lipman in the early 1970s at Montclair University (Montclair, NJ, USA). Oyler [1] claimed that the Montclair University (New Jersey) Institute for the Advancement of Philosophy for Children (IAPC) has established partnerships with over 40 countries, whose national education programs employ this technique. P4C is not only a student-centered teaching approach, at the same time, it is community-centered, specifically during teaching and learning (T and L) [2]. P4C is important because children hunger for meaning and lack reasoning [3,4]. It aims to encourage critical, creative thinking, to increase self-confidence and to improve academic achievement [2]. The main objective of P4C is to strengthen students’ curiosity (inquiry), help students make their own reasonable judgements and allow them to judge questions and issues philosophically (i.e., to display philosophical judgement) based on their own experience by involving other skills, namely critical thinking, creativity and “caring thinking” [1].

The basics of a P4C lesson include reading an episode from Lipman’s book followed by questions raised by students and discussion by the group on their chosen topic. The teacher might extend this discussion with questions from a discussion plan, or a prepared exercise exploring a particular philosophical issue [4]. One problem which arises for teachers is the lack of literary style in Lipman’s book. This leads others to seek philosophical stimulus from the traditional theories, picture books and from other kinds of writing. Fisher (2013) [4] added that a P4C session should start with a group of children or adults sitting in a circle of chairs, with the teacher as part of the group. The students read the book aloud round the class. After that, the teacher asks the students to pick out what they find strange, interesting, puzzling or worthy of discussion in the story. There is thinking time and time to share the discussion points that arise. Then, the students or teacher write on the wallboard, adding the name of the contributor as a sign of ownership of the questions. When there is a sufficient number of suggested topics, the teacher encourages the group to decide which topics will be the focus of the discussion. Then, the students discuss the topic. To extend the main ideas and thinking skills relating to the story, the students will answer the questions around the central concept or problem.

Many studies have been undertaken on the effectiveness of P4C programs in improving thinking skills among students [2,5,6,7]. For instance, in Malaysia, applying P4C in the classroom could help students in improving critical thinking. Research by Zulkifli and Hashim [5] revealed that P4C had helped to improve students’ critical thinking. The study was conducted via quasi-experimental research comprising 27 students placed in the experimental group while the remaining 34 students were placed in the control group. Based on the statistical t-test on critical thinking, the treatment group scored a higher post-test mean score compared to the control group. Furthermore, P4C helps students to be able to produce questions from lower order thinking questions (LOTs) to higher order thinking questions (HOTs) according to the hierarchy of Bloom’s taxonomy [6]. Next, P4C was found to help increase the teachers’ effectiveness in the classroom in terms of their ability to engage in dialogic and inquiry teaching, to develop relevant curricular materials for such teaching, to reflect on their own teaching, to recognize their students’ capacity for constructing knowledge through dialogic inquiry, to transfer the teaching strategies learned in P4C to other non-P4C lessons, and to identify and analyze philosophical concepts in the school curriculum. The findings of this study suggest that P4C plays a significant role in promoting the professional development of teachers [7].

Although these studies have been proven using various empirical data from around the world, the effectiveness of P4C should be seen holistically, not only in terms of the thinking (cognitive) skills of students but also from various other positive angles. Therefore, this SLR study aims to identify the extent to which P4C has positive effects for students, other than developing their thinking skills. P4C has been implemented by over 60 countries and many literature reviews have been conducted, but the systematic literature review (SLR) methodology has remained poorly implemented. The number of SLR studies is still not encouraging, while the scopes of the existing studies have still not been sufficiently broad. An SLR study was conducted through a quasi-experimental method on school students in England over 40 years. The study’s findings showed that P4C positively affected the students’ reasoning, reading and non-cognitive skills [8,9]. Meanwhile, the second SLR study by Ventista and Paparoussi [10] focused only on experimental and quasi-experimental studies, with the study supporting the cause-and-effect relationship between the P4C intervention and its findings.

Therefore, it can be concluded that SLR studies regarding the P4C program have been given a little attention. Researchers need to conduct SLR studies because it has been highlighted that the traditional literature contains various issues related to transparency, author bias, recruitment bias and publication bias. The SLR approach is needed because it is a comprehensive, transparent, structured and systematic literature-highlighting technique. This SLR has been guided by the main research questions, which explore the extent to which the community of inquiry (COI) in the P4C program is effective in having a positive impact on students, as well as the challenges that teachers and students encounter when the P4C program is being implemented. 

The main research goals of this SLR are, first, to explore the impact of the P4C program in the classroom, second, to explore the challenges faced by teachers using P4C in the classroom, and third, to explore the challenges faced by students using P4C. This research is of particular benefit to beginners or those who are not familiar with P4C, enabling them to engage with the overall ideas on the impact and challenges in implementing P4C in the classroom.

## 2. Materials and Methods

This review protocol was registered on the International Prospective Register of Systematic Reviews (PROSPERO).This SLR was created with reference to PRISMA (preferred reporting items for systematic review and meta-analysis), an extensively utilized standard resource in the medical and public health fields. PRISMA comprises 27 items as part of the SLR process [11]. Although this SLR was conducted in a social science field, PRISMA was still appropriate as it helped to form clear research questions and allowed systematic searches to be performed. In addition, PRISMA can minimize various types of bias and help authors to synthesize a study effectively [12].

The initial step in forming this SLR was to formulate appropriate research questions. The two main focus areas of this SLR were to explore the effectiveness of P4C, as well as the issues encountered by teaching staff and learners while it was being implemented. These themes formed the basis for the following research questions:What is the impact of the P4C program in the classroom?What are the challenges faced by teacher using P4C?What are the challenges faced by students using P4C?

### 2.1. Systematic Search Strategy

#### 2.1.1. Identification

Suitable keywords to use in the process of searching for articles and references for an SLR are defined and diversified through identification. Any process of searching requires keywords, which enable SLR references and articles to be more accurate in terms of the topics located. Five main keywords were selected based on the research questions: philosophy for children, philosophy with children, P4C, philosophical inquiry, active learning, creative thinking, critical thinking and teaching. The search for the primary keywords was diversified using terms with the same and similar meanings, as well as variants of the keywords. An online thesaurus was employed for the search process, while the researchers referred to keywords used in previous research studies, the Scopus database and experts’ opinions.

Following the selection of keywords, two principal databases were employed in the process of searching for articles and references, Web of Science (WoS) and Scopus. Their various strong points were the reasons for selecting these databases. As research by Martín-Martín et al. [13] confirmed, WoS and Scopus offer certain benefits, such as controlling the quality of results and indexing systematically. Within each database (WoS and Scopus), articles and references were located using advanced searching techniques but basic functions, for instance, Boolean operators (such as AND and OR), searches for phrases, truncated terms, wildcards and field codes. Overall, 151 articles and references were identified in Scopus and 140 were found in WoS using the search strategies applied with these keywords and databases. As Table 1 shows, the articles and references then underwent screening, the secondary phase of all systematic search strategies.

#### 2.1.2. Screening

The process of screening was applied to the 291 articles that had been gathered through the identification stage. The screening phase involves establishing inclusion and exclusion criteria, from which the final selection will be made up of suitable articles and references to be used in the SLR [14]. The first criterion used in this SLR was the year of publication, in which publications in the last five years (2016 to 2020) were selected. The selection of this period was based on several justifications. Firstly, it aligns with the work of Ishak et al. [15], who explored the theme of a study’s maturity. During this period, many related articles were successfully obtained. To ensure quality, only certain journal articles were selected for the review. Furthermore, to facilitate understanding, the published articles had to be written in English in order to be considered for selection.

Next, only articles with empirical data were selected, so review articles were excluded. This is because the main objectives of an SLR are to identify and understand the findings of past studies and not reviews of past studies. Another inclusion criterion used was the focus of the findings. The articles selected should present findings on the implementation of the P4C program involving students, involve a sample of the population, as well as qualitative methods, quantitative methods or mixed methods in the research study design. Journal articles that do not have complete data were removed. This was crucial since it enabled every article chosen to outline results that were related to the current SLR (refer to Table 2).

After the screening process, 133 articles were removed as they did not meet the set criteria. Hence, only 55 were retained for the next process.

#### 2.1.3. Eligibility

All the selected articles underwent a second screening process. This process is known as eligibility. Eligibility was determined to ensure that all the selected articles were relevant to the topic and could be included in this SLR. The chosen articles’ titles and abstracts were referred to as part of this process. If the information in the title and abstract appeared insufficient, then the methodology, results and discussion sections were reviewed. Another 40 articles were removed through this process because their discussions did not focus on the P4C program. If the articles did not present empirical data or they were duplicated records or in the form of scoping reviews, they were removed. After this process, 15 articles were selected for the next stage of the process, the quality assessment (see Figure 1).

#### 2.1.4. Article/Reference Quality Evaluation

A quality evaluation was performed on the articles and references chosen. This process is important to minimize bias and detect articles that may contain weaknesses in terms of methodology [17]. Assessors were selected among the researchers for this assessment. As the articles and references from different and combined types of studies—quantitative, qualitative and mixed methods—can be used in SLRs, MMAT (mixed methods appraisal tools) were used to help the evaluators when they conducted the evaluation [18]. According to the study designs used in the articles and references, two basic criteria and five specific criteria were used to evaluate the selected materials. The two basic criteria were used to evaluate the articles’ quality in the initial stage of this procedure. These were, first, whether the outlined research questions were clear and, second, if the information provided was able to address the given research questions. Each of these conditions must be met before an evaluated article could proceed to the following phase, in which it would be categorized based on the study design—qualitative, quantitative or mixed methods. Subsequently, five specific criteria were used to evaluate the remaining articles.

For each criterion, the evaluator answered based on three choices, namely Yes (Y) or No (N), while if they were unclear or not sure about the results of the evaluation, they would choose the answer Cannot tell the result (Cl). To evaluate each article, both evaluators had to come to a mutual agreement for each evaluation. If an agreement was not reached, they were given a third opinion. Only articles/references that met at least three of the five criteria were deemed quality articles and included in the SLR. Out of the 15 articles evaluated, nine articles met at least three inclusion criteria for the SLR. Meanwhile, four articles—those by Cassidy, Conrad et al. [19], Çayır [20], Curko and Cah [21], and Malboeuf-Hurtubise et al. [22]—were removed as they did not meet the set minimum criteria. Two more articles were removed—namely those by Duytschaever et al. [23] and Hawken [24]—as they were unrelated to the topic (refer to Table 3). 

#### 2.1.5. Data Extraction and Analysis

Using the articles chosen, the following phase was the extraction of data. Two researchers undertook this procedure. The process of extracting the data focused on three key elements of the articles: the abstracts, the study findings and the discussion. The reason was that the current SLR focused on reviewing how effective the P4C program was, according to the results of past research. Different parts of the article were read if they appeared to contain useful information. To enable the subsequent process of analysis to be conducted more easily, the data extracted was tabulated. Once the relevant data had been extracted, the next step was to perform data analysis. Since this SLR is an integrative review that combines various study designs (quantitative + qualitative + mixed methods), the qualitative synthesis was considered the best analysis [25]. Various analyses can be used in a qualitative synthesis. Thematic analysis, as Flemming et al. [26] argued, is one of the optimal methods of qualitative synthesis to use when conducting analyses of the results from different study design approaches. Utilizing similarities or relevant aspects of the study results that have been gathered, thematic analysis tries to locate patterns in the existing research. In order to find a suitable theme, the findings that had been extracted were examined one by one, and if findings had similarities or relevance, they were placed in one data set. A suitable theme was then allocated to each of these data sets. There were five themes were identified, namely (1) higher-order thinking skills, (2) a safe classroom environment, (3) civilized students, (4) democracy in discussion, and (5) the culture of thinking in the classroom (refer to Table 4).

#### 2.1.6. Articles Background

Before commenting on the key findings, this section will focus on the background of the articles/references selected for the SLR. Out of the 15 articles/references selected, two were published in 2020, two were published in 2018, three were published in 2017 and two articles/references were published in 2016 (refer to Table 3). The selected articles were published in different journals, namely the *Journal of Childhood Philosophy*, the *Journal of Educational Review*, the *International Journal of Instruction*, the *Journal of the Reading Association of South Africa*, the *Journal of Frontiers in Psychology*, the *Journal of Philosophy of Education*, the *Journal of Emotional and Behavioral Difficulties*, the *Journal of Social Work* and the *Journal of Education, Citizenship and Social Justice*.

## 3. Results

### 3.1. Theme 1: Higher Order Thinking Skills

The findings of the SLR study in terms of the effectiveness of the P4C program and the challenges faced by teachers in the implementation of P4C are referred to in Table 5. One of the main themes formed in this SLR was higher-order thinking skills (HOTS). The first sub-theme showing the effectiveness of the P4C program and the inquiry community was reasoning skills. Students will discuss an issue, build arguments and explain using examples to strengthen their arguments (articulation). Students dare to explain their views without feeling that their views will provoke controversy. In addition, teachers will also encourage learners to express their views by providing strong reasons, along with examples to support their views [9,16,27,28,29].

Moreover, the second and third sub-themes are critical and creative skills [27,28,30,31]. In COI, students are allowed to practice both of these skills. They learn to be focused listeners and then seek other perspectives from different participants. Students will ask to confirm that the information is accurate before constructing their thinking. Eventually, they will argue by creating new views based on the information obtained during the discussion [27]. Similarly, among students who are unable to read and write, according to Murris and Thompson [31], their critical and creative skills can be seen through the drawings they produce during the sessions.

The fourth sub-theme is the skill of constructing and making connections (analogy). It is one of the effective aspects of COI for students, whereby during discussions in COI, students can make connections with an event to support their views and provide explanations during discussions. Students usually connect their own experiences to explain a situation [27,28]. Students who cannot read and write will draw connections based on their life experiences [31]. In the meantime, the fifth sub-theme is concept-building skills and criteria, which refer to students’ ability to define a concept, such as explaining something and supporting the concept with other examples [9]. The sixth sub-theme is constructing questions at the beginning of the discussion session. In P4C, the skill of constructing philosophical questions is fundamental in a discussion because constructing open-ended questions (philosophical question) stimulates students to continue to discuss and hold dialogue [9,29]. The seventh sub-theme is decision-making skills. While in COI, students are not in a hurry to express their views on an issue. They will share their views when they think that these views are appropriate. This change in students’ attitudes is visible when they are engaged in discussions in COI [10,28]. The eighth sub-theme is evaluating skills. In this regard, students understand that not all questions have correct answers; therefore, they can evaluate and choose the appropriate answer for themselves during the discussion [29]. Assessment skills require a strong degree of concentration before a student agrees or disagrees with something before giving their views [10]. The ninth and the last sub-theme is reflective skills. During a discussion, students must think deeply before expressing their opinion [20,28]. Students are also more open to accepting other views as answer choices without regarding views different from their own as criticism [29].

### 3.2. Theme 2: Democracy

The second theme is democracy. There are three sub-themes under this theme. The first is that students are free from being labelled (free from judgmental behavior) while giving different views. During sessions in COI, students feel confident to provide insights and are not worried about being seen differently or labelled with a particular title by the adults or teachers who are with them [28,31]. The second sub-theme is mutual respect and tolerance when providing views in COI. Each student can explain their views in an organized and orderly manner, even if they disagree. All the students accept each view openly [29], despite their differences of opinion. The third sub-theme is the search for social justice. A discussion in COI becomes more in-depth when students begin to empathize with marginalized groups, including oppressed women, underprivileged children and other such groups [28].

### 3.3. Theme 3: Civilised Students

The third theme is the effectiveness of the P4C program in producing civilized students. Students who were initially difficult to manage have started to follow the rules such as sitting in an orderly manner, being active listeners, being patient before expressing views and not being rude when arguing with teachers or their peers [9,16,27,29,30]. 

### 3.4. Theme 4: Culture of Thinking

The fourth theme is that the COI session can create a culture of thinking in schools. According to Gorard et al. [30], P4C has created a new school culture that consists of thinking, listening, talking and making arguments in schools. 

### 3.5. Theme 5: Safe Environment

The fifth theme is that P4C has made the classroom an intellectually safe environment for the development of learning. COI makes the learning environment a free place for students to socialize and relieve emotional stress [28].

### 3.6. Challenges Faced by Teachers

There were three challenges faced by teachers highlighted in the reviewed articles, namely optional methods, classroom management and deficiency of ideas (Table 6). First, one of the challenges faced by teachers is that the P4C program is optional as a method or module to be employed in schools. It is not part of the national curriculum [8]. In addition, the key concept in the P4C discussion emphasizes knowledge and belief. For teachers, an in-depth discussion of this concept is considered unimportant because it is not part of the subject content [29]. In addition, teachers need to be prepared in terms of ideas and resources with which to plan their lessons [30]. The second challenge is classroom management, which consists of controlling the classroom, time constraints, paying attention to each student equally, and meeting the students’ psychological and social needs. Teachers need to be fair during discussions and honor all the students when they give their views. For example, teachers need to allow time for students to think before answering and pay attention while students give insights. Teachers need to avoid bias in discussion sessions [29,30,32]. For time constraints, as teachers are also involved with other school activities, they do not have much time to complete the syllabus [29,30]. Sometimes teachers are not paying attention to all students [30] and cannot meet students’ psychological and social needs. This causes students to feel that a lack of intimacy (engagement) exists between them and causes them to be unmotivated [29]. The final challenge for teachers was the deficiency of ideas that provide stimulating materials, causing students to feel bored [30].

### 3.7. Challenges Faced by Students

There were three challenges faced by students, including lack of interpersonal skills (Table 7) [28], and various learning problems such as autism, causing students to be slow in organizing and communicating information. Therefore, they need the assistance of teachers when parsing views (inference) [16]. Lack of knowledge was also a challenge. Lack of knowledge involved lack of vocabulary among students. This causes them to face problems when giving their arguments [27] and causes students to feel unappreciated. Teachers also need to be more careful when teaching students with specials needs [27]. Furthermore, some of the students were not happy with P4C because they were reticent and reluctant to share their views with others [29].

## 4. Discussion

Overall, the findings showed that there were five themes related to the impact of P4C in the classroom, namely higher order thinking skills, democracy, civilized students, culture of thinking and safe environments. There were also three challenges for teachers in implementing P4C, such as optional methods, classroom control and deficiency of ideas, while the three challenges for students using P4C were lack of interpersonal skills, learning problems and lack of knowledge.

The impact of P4C was higher order thinking skills. It is in line with the main goal of the P4C program, which is to improve thinking skills [33] including critical thinking, creative thinking and caring thinking [33,34,35]. Caring thinking is a form of thinking that encourages a person to value other people’s opinions and to empathize with others. According to Lipman [36], one should consider the impact of one’s opinion or decision on others. Sensitivity to the views of others is part of caring for other people [37]. In addition, Ventista and Paparoussi [9] stated that P4C is said to be successful when students can follow the established rules by using the expressions I agree/I disagree before giving an opinion [9,35] and meaningful learning [34]. Research by Lam [7] supported that, P4C make students capable of reasoning and arguing in a competent way about philosophical problems arising from various stimuli prepared by their teachers. P4C was also found to play a major role in promoting the students’ critical and creative thinking and to enhance the development of their English language proficiency to a significant extent. Wu [38] also agreed that P4C has an impact of improvement in thinking skills which measured using a composite of validated critical thinking tests.

Next, the main challenge for teachers using P4C is classroom management. Abdul Rahman and Noor [39] claim in their study that if many activities are carried out in the classroom, then teachers find it difficult to control the classroom. Establishing rules and reinforcing consequences for misbehavior were the main approaches to classroom management, although more contact with actual classrooms and learning from experienced others were alternatives for improving classroom management skills [40].

Moreover, for students, the challenges of using P4C included the lack of interpersonal skills. ‘Interpersonal skill’ is one of a number of broadly similar terms that are sometimes used interchangeably. Other such terms include interactive skills, people skills, face-to-face skills, social skills and social competence. A common theme in these definitions is the ability to behave in ways that increase the probability of achieving desired outcomes. It therefore seems appropriate to define interpersonal skills as goal-directed behaviors used in face-to-face interactions in order to bring about a desired state of affairs [41]. There are many interpersonal challenges that children confront in school such as they are typically faced with shifting social ecologies, relationships, and resources. Beyond basic tasks such as relating with classmates and forming ties with teachers, children find that they are under increasing pressure to compare and evaluate themselves, their abilities, and their achievements to those of their agemates [42].

The research suggested that there should be class time dedicated to P4C to overcome the challenges faced by teachers and students. Teachers should also receive more training, especially in their role as facilitators and in the methods for conducting philosophical discussion. 

Furthermore, safe environment themes are the new findings on the impact of P4C because P4C creates environments in which students are not afraid of giving their opinions and can give reasons for their ideas.

## 5. Conclusions

P4C is a productive pedagogical approach. According to Luke and Hogan [43], a productive pedagogy builds relationships between pedagogy and knowledge by integrating the dimensions of intellectual quality, relevance, and a conducive classroom environment, as well as appreciating and addressing differences holistically in the teaching and learning processes [15]. The suggestion for future researchers is to provide teaching modules using P4C to facilitate their implementation by teachers in the classroom. The construction of stimulating module materials requires cooperation between teachers because the issues and ideas that need to be brought to the students must be appropriate to the students’ cultural environment, religion and level of thinking.

## Figures and Tables

**Figure 1 children-09-01671-f001:**
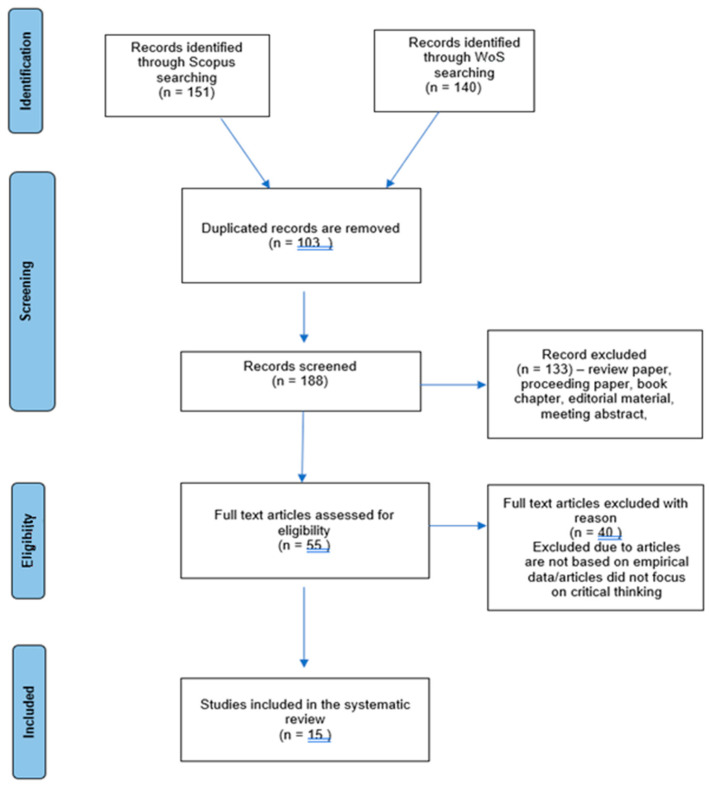
Flow diagram of the study (adapted from Moher et al. [16]).

**Table 1 children-09-01671-t001:** The search strings.

Database Search String	
WoS (*n* = 140)	TS = (“p4c” OR “philosophical inquiry *” OR “philosophy with children” OR “philosophical with children” OR “active learning” OR “creative thinking” OR “critical thinking”) AND “teaching”)
Scopus (*n* = 151)	TITLE-ABS-KEY (“p4c” OR “philosophical inquiry *” OR “philosophical with children” OR “philosophical with children” OR “active learning” OR “creative thinking” OR “critical thinking”) AND “teaching”)

* means different forms of a word.

**Table 2 children-09-01671-t002:** Inclusion criteria used.

Inclusion Criteria	
Year of publication	6 years (2016 to 2021)
Publication type	Journal Articles
Type of language	English
Type of findings	Empirical
Main findings	Data related to application and effectiveness of P4C program

**Table 3 children-09-01671-t003:** Selection of basic study criteria.

Article/ Reference	Cassidy, Christie et al. [4]	Cassidy, Conrad et al. [13]	Cassidy, and Heron [3]	Cassidy, Marwick et al. [5]	Çayır [14]	Curko and Cah [15]	Duytschaever et al. [18]	Gorard et al. [6]	Hawken [19]	Leng [20]	Malboeuf-Hurtubise et al. [17]	Rahdar et al. [21]	Siddiqui et al. [22]	Murris and Thompson [23]	Ventista and Paparoussi [10]
**Basic criteria/study**															
Are the data obtained able to answer the research questions stated?	Y	Y	Y		N	N				Y				Y	Y
**Qualitative Criteria**															
Is the qualitative technique employed suitable for answering the research questions?	Y	Y	Y		N	N				Y				Y	Y
Is the qualitative data collection methodology used adequate for answering the research questions?	Y	Y	Y							Y				Y	Y
Are the results of the study obtained from the data sufficient?	Y	N	Y		N	N				Y				Y	Y
Can the interpretation of the study results be substantiated with data?	Y	Y	Y		N	N				Y				Y	Y
Do the sourcing, collecting, analysing and interpretating of the qualitative data display continuity?	Y	C	Y		N	N				Y				Y	Y
**Results**	A	R	A		R	R				A				A	A
**Basic criteria/study**															
Are the research questions clearly stated?											Y	Y	N		
Are the data obtained able to answer the research questions stated?											C	Y	N		
**Quantitative Criteria**															
Is the sampling strategy used sufficiently relevant to answer the research questions?											N	Y	Y		
Is the studied population represented in the chosen sample?											N	Y	Y		
Are the measurements used appropriate?											Y	Y	Y		
Is there a low risk of biased non-response?											Y	Y	Y		
Are the research questions answered using a suitable form of statistical analysis?											Y	Y	Y		
**Results**											R	A	A		
**Basic criteria/study**															
Are the research questions clearly stated?				Y				Y							
Are the data obtained able to answer the research questions stated?				Y				Y							
**Mixed methods study**															
Is there a reason for using mixed methods to answer research questions?				Y				Y							
Can these different research components be combined effectively to answer the research questions?				Y				Y							
Are the combined qualitative and quantitative results interpreted accurately?				Y				Y							
Did the authors sufficiently address any differences or inconsistent aspects involving the quantitative and qualitative findings?				Y				Y							
Do the different study components comply with the quality criteria for each study design involved?				Y				Y							
**Results**				A				A							

Y = Yes; N = No; C = Cannot tell; A = Accepted; R = Reject.

**Table 4 children-09-01671-t004:** The matrix of themes and the sub-themes of impact P4C in the classroom.

Themes	Higher-Order Thinking Skills (HOTS)									Democracy			Civilised	Culture of Thinking	A Safe Class Environment
Sub-Themes	CRV	RSN	CRT	ANA	CNP	QST	RST	EVL	RFT	LBL	RES	SCE			
Cassidy, Marwick et al. [5]		x						x	x				x		
Cassidy and Heron [3]	x	x	x	x				x					x		
Cassidy, Christie et al. [4]		x							x						
Leng [20]		x	x	x			x		x	x		x			x
Rahdar et al. [21]			x												
Siddiqui et al. [22]	x	x				x		x	x		x		x	x	
Murris and Thompson [23]	x			x						x					
Ventista and Paparoussi [10]		x			x	x							x		
Gorard et al. [6]			x												

CRV: Creative, RSN: Reasoning, CRT: Critical, ANA: Analogy, CNP: Concept, QST: Question, RST: Result, EVL: Evaluation, RFT: Reflective, LBL: Judgmental/Labelling, RES: Respect, SCE: Social Equality.

**Table 5 children-09-01671-t005:** Themes and Subthemes of the impact of P4C in classroom.

Research Questions	Themes	Subthemes
What are the impacts of the P4C in the classroom?	Higher Order Thinking Skills	Reasoning Skills
		Critical Skills
		Creative Skills
		Analogy
		Concepts
		Questions
		Decision Making
		Evaluating Skills
		Reflective
	2. Democracy	Free from being labelled
		Mutual respect
		Social Justice
	3. Civilized Students	
	4. Culture of Thinking	
	5. Safe Environment	

**Table 6 children-09-01671-t006:** Challenges faced by teachers highlighted in the reviewed articles.

Challenges Faces by Teachers
Optional methods [8,29,30]
Classroom managements [29,30,32]
Deficiency of Ideas [30]

**Table 7 children-09-01671-t007:** Challenges faced by students highlighted in the reviewed articles.

Challenges Faces by Students
Lack of interpersonal skills [28]
Learning problem [16]
Lack of knowledge [27]

## Data Availability

Not applicable.

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
