# Peer review of "Impact of Philosophy for Children and Its Challenges: A Systematic Review"

_children, 2022, doi:10.3390/children9111671_

Round 1

Reviewer 1 Report

Thank you for the Author's contribution to this manuscript. I would like to give some feedback to improve your manuscript. 

·      The introduction of this manuscript is short. The main things that I missed are the detailed importance (why is important) and methods of P4C.

·      Please add the main research goal to the end of the introduction. 

·      The description of the Material and methods are extremely detailed. It is the strong point of this study. It is easy to follow and contains all the necessary information that the reader needed. 

·      I recommend adding a new table for the results where the Authors could highlight the main finding of the included studies. 

·      There is no table 5. (Line 269)

·      I recommend to highlighted once again all the important findings in the discussion part. 

·      It also would be suitable if the Authors would make suggestions based on their findings.           

Author Response

i had done the correction

Reviewer 2 Report

As the authors note, this program has been in existence for 50 years and literally thousands of studies have been done on this pedagogy. In light of that, it is not clear that this article adds anything new in terms of its findings. On the other hand, the fact that it is a carefully done SLR is relatively novel, as is the fact that it seeks to publish in a journal that may be frequented by those who are not familiar with P4C.

The fact that the SLR was restricted to papers published in the last 5 years has resulted in an odd grouping, i.e., there are many older excellent studies that are noticeably absent. As well, the summary of the results, while they may pique the interest to those not familiar with P4C, will seem rather superficial for those who intimately connected to the program. Thus, for instance, the impression is given that higher order thinking skills are an inevitable result of the program. Nowhere is there a comment on how long students have to be engaged in the program before such results emerge, and nor does it comment on what kind of training facilitators have undergone. It also notes that students are free from being labeled but neglects the fact that “politically correct forces” ensure that dog whistles can silently label—a fact that needs to be handled with finesse. In the challenges faced by the teachers, there is not even a hint of what kind of training they have undergone. And under challenges faced by the student, the comment that “teachers can give rewards if the student can give their opinion” is worrying as it suggests a COI is primarily about opinion gathering, which, if it were, would not have the positive results articulated in this article.

Line 398—“irrelevance” I think should be “relevance.”

My overall opinion is that this article will be of interest to those outside the world of P4C but not to those who are presently in the process of trying to make it ever better—one of the major problems being how to train facilitators so that the laudatory results of which these authors speak in fact materialize.

Author Response

I had done the correction.

Round 2

Reviewer 1 Report

Thank you for the author's contribution. The manuscript was significantly improved.